# Kidney Function Change and All-Cause Mortality in Denosumab Users with and without Chronic Kidney Disease

**DOI:** 10.3390/jpm12020185

**Published:** 2022-01-31

**Authors:** Ping-Hsun Wu, Ming-Yen Lin, Teng-Hui Huang, Tien-Ching Lee, Sung-Yen Lin, Chung-Hwan Chen, Mei-Chuan Kuo, Yi-Wen Chiu, Jer-Ming Chang, Shang-Jyh Hwang

**Affiliations:** 1Department of Internal Medicine, Division of Nephrology, Kaohsiung Medical University Hospital, Kaohsiung 80756, Taiwan; 970392@kmuh.org.tw (P.-H.W.); mingyenlin3@gmail.com (M.-Y.L.); sfestg3329@gmail.com (T.-H.H.); mechku@kmu.edu.tw (M.-C.K.); chiuyiwen@kmu.edu.tw (Y.-W.C.); jemich@kmu.edu.tw (J.-M.C.); sjhwang@kmu.edu.tw (S.-J.H.); 2Faculty of Medicine, College of Medicine, Kaohsiung Medical University, Kaohsiung 80708, Taiwan; 3Department of Orthopedics, Kaohsiung Medical University Hospital, Kaohsiung Medical University, Kaohsiung 80756, Taiwan; tony8501031@gmail.com (S.-Y.L.); hwan@kmu.edu.tw (C.-H.C.); 4Orthopaedic Research Center, College of Medicine, Kaohsiung Medical University Hospital, Kaohsiung Medical University, Kaohsiung 80708, Taiwan; 5Regenerative Medicine and Cell Therapy Research Center, Kaohsiung Medical University, Kaohsiung 80708, Taiwan; 6Department of Orthopedics, College of Medicine, Kaohsiung Medical University, Kaohsiung 80708, Taiwan; 7Department of Orthopedics, Kaohsiung Municipal Ta-Tung Hospital, Kaohsiung 80145, Taiwan

**Keywords:** denosumab, adherence, mortality, renal function, chronic kidney disease

## Abstract

Denosumab is approved for osteoporosis treatment in subjects with and without chronic kidney disease (CKD). Confirmation is required for its safety, treatment adherence, renal function effect, and mortality in patients with CKD. A retrospective cohort study was conducted to compare new users of denosumab in terms of their two-year drug adherence in all participants (overall cohort) and CKD participants (CKD subcohort), which was defined as baseline estimated glomerular filtration rate (eGFR) less than 60 mL/min/1.73 m^2^. The eGFR was calculated using the 2021 CKD-EPI (Chronic Kidney Disease Epidemiology Collaboration) equation. We defined high adherence (HA) users as receiving three or four doses and low adherence (LA) users as receiving one or two doses. All-cause mortality was analyzed using Kaplan–Meier curves and Cox regression models. In total, there were 1142 subjects in the overall cohort and 500 subjects in the CKD subcohort. HA users had better renal function status at baseline than LD users in the overall cohort. A decline in renal function was only observed among LD users in the overall cohort. In the CKD subcohort, no baseline renal function difference or renal function decline was demonstrated. The all-cause mortality rate of HA users was lower than LA users in both the overall cohort and CKD. A randomized control trial is warranted to target this unique population to confirm our observations.

## 1. Introduction

Chronic kidney disease (CKD) is commonly associated with mineral and bone disorders, osteoporosis, and an increased risk of fractures [1,2,3,4,5]. The World Health Organization defines osteoporosis as being diagnosed with a T-score of ≤−2.5. CKD is an independent risk factor for osteoporosis [6] and fragility fractures, especially hip fractures, are associated with high complications and mortality rates [7,8,9]. Eighty-five percent of women with osteoporosis have mild-to-moderate renal impairment [10]. Since osteoporosis and CKD have a strong association, it is important to treat osteoporotic patients with renal insufficiency effectively and safely, without causing any adverse effects on intrinsic renal function [5].

Denosumab (Prolia^®^; Amgen, Thousand Oaks, CA, USA) is a human monoclonal antibody that targets the receptor activator of the nuclear factor kappa-B ligand, which inhibits osteoclasts and increases bone mineral density (BMD) [11]. It is not metabolized or excreted by the kidney, in contrast with other antiresorptive medications, such as bisphosphonates. Denosumab has been authorized for anti-osteoporosis treatment in Taiwan since 2011 and has been covered under the insurance program of the National Health Insurance (NHI) benefits since 2012. The three-year, pivotal, three-phase placebo-controlled FREEDOM trial in postmenopausal women with osteoporosis proved the efficacy and safety of denosumab in patients with and without renal impairment [5,12].

Osteoporosis is a chronic disease that requires long-term treatment. Adherence to the anti-osteoporotic medication has been challenging, regardless of the dosing interval or route of administration [13]. Discontinuation of denosumab use may result in a rapid decline in BMD and reversal of the inhibition of bone remodeling [14,15]. Denosumab’s 24-month persistence rate was reported to be 40–86%, declining with time [16,17,18,19,20,21,22]. Since research regarding denosumab adherence in patients with CKD is limited, this study was designed to evaluate denosumab adherence in terms of change in renal function among osteoporotic patients with and without CKD. We also investigated the impact of drug adherence on mortality rates.

## 2. Materials and Methods

### 2.1. Database

This retrospective cohort study used data from the Kaohsiung Medical University Hospital (KMUH) Research Database (KMUHRD) of KMUH, a medical center with 1600 beds, which received approximately 6000 clinical visits per day in 2015. The KMUHRD provides comprehensive data on hospital care, ambulatory care, biochemical data, and drug dispensing records. All diagnoses are recorded according to the International Classification of Diseases, 9th Revision, Clinical Modification (ICD-9-CM) or International Classification of Diseases, 10th Revision, Clinical Modification (ICD-10-CM). Drug dispensing data includes the type of prescriber, the name, date, amount, and prescribed dose regimen of the dispensed drug, and the length of the prescription (drug treatment period). The KMUHRD is managed by the Division of Medical Statistics and Bioinformatics of KMUH. For confidentiality, and according to the Personal Information Protection Act, all personal identifiers are removed, and only authorized researchers are allowed to conduct data linkage, processing, and statistical analysis. All-cause mortality was validated through the national death registry. The cause of death was identified according to ICD-10-CM.

This study was approved by the institutional review board of KMUH (KMUHIRB-E(I)-20210018), which waived the requirement of informed consent for this retrospective study. All clinical investigations were conducted according to the principles of the Declaration of Helsinki.

### 2.2. Study Subjects, Comorbidities, and Medications

Patients with osteoporosis and a denosumab prescription were enrolled from 1 January 2012 to 31 December 2017. The index date was defined as the date of initiation of denosumab treatment. We excluded patients with incomplete demographic data or who were receiving dialysis treatment, taking other osteoporosis drugs, had a malignancy diagnosis with denosumab treatment, or who died within two years after the index date. New users of denosumab for osteoporosis were stratified into two groups according to their drug adherence. The high-adherence (HA) group was defined as patients who had received more than two denosumab injections (receiving 3 or 4 doses) at 24 months after the index date. The low-adherence (LA) group comprised all other patients (receiving 1 or 2 doses).

The baseline data recorded were age, gender, history of fracture, Charlson comorbidity index (CCI), and medications. Deyo’s CCI was defined according to ICD-9-CM or ICD-10-CM coding (Appendix A) [23]. Other diagnostic information is shown in Appendix A. Medications were classified according to the Anatomical Therapeutic Chemical classification (Appendix A). Cardiovascular and noncardiovascular mortality were defined according to the ICD-10-CM classification (Appendix A). The estimated glomerular filtration rate (eGFR) was calculated using the equation of the 2021 CKD-EPI (Chronic Kidney Disease Epidemiology Collaboration). CKD was defined as eGFR < 60 mL/min/1.73 m^2^.

### 2.3. Statistical Analysis

Our data are presented as percentages for categorical variables or means ± standard deviation for continuous variables. Between-group differences were analyzed using the Student’s *t*-test for independent continuous variables, paired *t*-test for dependent continuous variables, and the Chi-squared test for categorical variables. The Kaplan–Meier method was used to determine survival curves for all-cause mortality, cardiovascular mortality, and noncardiovascular mortality. The association between denosumab treatment adherence and mortality was assessed using univariate and multivariate Cox proportional hazard analyses. Their outcomes are presented as hazard ratios (HRs) with 95% confidence intervals (CIs). Age, gender, and CCI were used to adjust for confounders in the multivariate analysis. A *p*-value of <0.05 indicated a statistically significant difference. Data processing and statistical analyses were performed using SAS ver. 9.4 (SAS Institute, Cary, NC, USA).

## 3. Results

We enrolled 1142 new users of denosumab for osteoporosis treatment after excluding subjects with incomplete demographic data (*n* = 3), those undergoing dialysis treatment (*n* = 37), with exposure to other osteoporosis drugs (*n* = 930), malignancy diagnosis with denosumab treatment at baseline (*n* = 146), and who died within two years after the index date (*n* = 280). Subjects without renal function records were also excluded to enable the evaluation of change in eGFR. We compared new users of denosumab (HA [*n* = 713] and LA [*n* = 409]) to evaluate the differences in renal function change and long-term all-cause mortality. To assess the impact of CKD on denosumab treatment, a CKD subcohort was evaluated on the same terms (Figure 1).

Comparisons of the baseline characteristics between HA and LA denosumab users were performed within the overall cohort and the CKD subcohort. In general, there were no differences in age, gender, overall fracture history, CCI score, or concurrent medication between the two groups in either the overall cohort or the CKD subcohort. The HA group had a higher proportion of non-hip fractures than the LA group. The HA group was prescribed more NSAID treatments than the LA group. The HA group had less antihypertensive treatment than the LA group (Table 1). HA users had higher eGFR levels before and after the initiation of treatment compared with the LA group in both the overall cohort and the CKD subcohort. In terms of one-year average eGFR, no significant renal function decline was observed after denosumab initiation (Table 2).

The single comorbidity in Deyo’s CCI is shown in Appendix A. Kaplan–Meier curves of all-cause mortality demonstrated better survival in the HA group than the LA group in both the overall cohort and the CKD subcohort (log-rank *p* < 0.01) (Figure 2). The HA group had lower noncardiovascular mortality than the LA group in the overall cohort (Appendix A) but not demonstrated in the CKD subcohort (Appendix A). In the Cox regression analysis adjusted for confounders (age, gender, and CCI score), HA users were associated with a lower risk of all-cause mortality compared with LA users, corresponding to an adjusted HR of 0.64 (CI, 0.48–0.86) in the overall cohort and an adjusted HR of 0.61 (CI, 0.43–0.87) in the CKD subcohort (Table 3). For cardiovascular and noncardiovascular mortality, HA users were associated with a lower risk of noncardiovascular mortality than LA users (HR, 0.66; CI, 0.47–0.91) in the overall cohort and a lower risk of cardiovascular mortality (HR, 0.46; CI, 0.21–0.98) in the CKD subcohort (Appendix A).

In order to evaluate the possible blood calcium level changes, the 3-month mean ionized calcium level was compared before and after denosumab treatment. No difference in ionized calcium level was observed among HA or LA users in the overall cohort or CKD subcohort (Appendix A). However, only 10 to 17 participants had paired calcium data, so the interpretation should be cautious in our study.

## 4. Discussion

We evaluated denosumab treatment adherence, renal function change, and all-cause mortality using Taiwanese medical data. Our results show that adherence to a two-year denosumab treatment plan is not detrimental to renal function in osteoporotic patients with or without CKD. HA users represented 63.5% of our overall cohort and 59.8% of our CKD subcohort; these rates are similar to those previously reported [16,17,18,19,20,21,22]. The HA group had a lower all-cause mortality rate than the LA group in both the overall cohort and the CKD subcohort. In terms of the cause of death, we observed lower noncardiovascular mortality amongst HA users in the overall cohort and lower cardiovascular mortality in the CKD subcohort.

Patients with CKD have a greater risk of osteoporosis and fractures [24,25]; diagnosis and treatment in this special population remains challenging due to complications such as mineral and bone disorder, which involves complex mechanisms causing abnormalities of bone and mineral metabolism. The 2017, updated guidelines of the “Kidney Disease: Improving Global Outcomes” report recommend that anti-osteoporosis treatment decisions for patients with CKD stages 3A–5D should consider biochemical abnormalities and the progression of CKD. A bone biopsy should be considered prior to anti-osteoporosis treatment in order to understand the underlying bone pathology and avoid adynamic bone disease after antiresorptive agent use [26,27,28]. In a large, randomized clinical trial, Broadwell et al. demonstrated the long-term safety and efficacy of denosumab for anti-osteoporosis treatment in postmenopausal patients with normal renal function (to CKD stage 3) and an absence of clear abnormalities in mineral metabolism [5]. In our two-year study, denosumab use had no negative impact on renal function, which included male patients with CKD. The evidence shows that denosumab is safe in patients with mild-to-moderate CKD.

The positive effects of long-term denosumab use in terms of BMD improvement and fracture risk reduction have been demonstrated in large clinical trials [5,11,21,29,30,31]. Adherence to and persistence with anti-osteoporotic medication is critical to reducing fracture risk. In terms of adherence, a theoretical advantage of denosumab is the convenience of its biannual subcutaneous administration. A 2018 review article by Morizio et al. showed that denosumab might have better adherence and cost-effectiveness compared with oral bisphosphonates [32]. However, denosumab is a reversible agent; discontinuation of treatment is associated with a decrease in BMD and an increased risk of fractures, reversing the benefits of treatment [21,33]. In our study population, only 63.5% had high adherence, which implies that more than one-third were at high risk of fractures. To improve adherence and persistence, Kobayashi et al. suggested some strategies, including dental care, combining medications to prevent complications, and educating patients on the benefits and the necessity of continuing treatment [22]. Further clinical trials are needed to clarify their effects on real-world adherence.

A 2019 Taiwanese nationwide cohort study demonstrated lower all-cause mortality in patients with good adherence to anti-osteoporosis medications, including bisphosphonates, calcitonin, raloxifene, and teriparatide [34]. However, the impact of adherence to denosumab treatment on mortality has been investigated only by a limited number of studies. Using Austrian national data, Behanova et al. reported that hip fracture patients treated with antiresorptive medications, including bisphosphonates and denosumab, had significantly longer survival times than those without such treatment. However, Cox regression analysis with a time-varying covariate did not show a statistically significant difference for patients treated with denosumab [35].

Our study showed that high adherence to denosumab treatment was associated with better survival. There is no evidence to explain the lower mortality of denosumab users with good adherence. However, the prevention of subsequent fractures is one possible reason [36]. A 2010 randomized, controlled trial showed similar incidences of pneumonia, cancer, and cardiovascular disease among patients treated with zoledronate or placebo. Mortality rates were lower in those treated with zoledronate compared with placebo. The authors proposed that treating osteoporosis might improve the ability of osteoporotic patients to cope with acute illness, possibly by maintaining their physiological condition [37]. Although the mechanism of denosumab differs from that of zoledronate, there have been common postulates to explain their potential positive effects on survival. Chen et al. found that denosumab may suppress the progression of coronary artery calcification and regress osseous calcification in patients with secondary hyperparathyroidism and osteoporosis due to dialysis [38]. This may explain the lower cardiovascular mortality in the CKD subcohort of our study. Additional studies are required to investigate denosumab’s mechanisms and effects in reducing mortality.

Our study has some limitations. First, it is difficult to confirm any causality between denosumab use and death in an observational study. Second, our subjects were mainly severely osteoporotic patients. Reimbursement for anti-osteoporosis medication under Taiwan’s NHI program requires a patient with a BMD T-score of ≤−2.5 with or without a history of fractures. Our study’s generalizability to non-severely osteoporotic subjects (e.g., with osteopenia) is limited. Furthermore, our study participants were from academic medical centers; the results might not generalize to other settings or patients with fewer comorbidities. Third, residual confounders—such as smoking, alcohol, and body mass index—were not controlled for in this study. The mineral bone biomarkers (e.g., pathology from bone biopsy, circulating bone turnover markers, vitamin D, and parathyroid hormone) were not part of routine care nor health insurance-reimbursed tests in the study setting. Limited sample sizes make it difficult to observe the clinical biochemistry abnormality. For example, hypocalcemia in the osteoporotic patients who received antiresorptive medications is a concern, especially in subjects with CKD. We evaluated the ionized calcium level before and after the denosumab used in our study. Although we observed a mild decrease in ionized calcium, no statistical significance was found due to the limited sample size. Finally, our analyses performed according to different denosumab drug adherence, but it is difficult to completely avoid an indication bias in observational studies that evaluate the effect of medication. Hence, the existence of unidentified residual confounders (i.e., the effect of the other exposures and potentially adverse effects) cannot be completely ruled out from the present study. For example, a comparison of HA and LA users and the risk of fracture events is straightforward. However, selection bias, confounding by indication, and unadjusted confounders should be considered for this approach. Severe osteoporosis is prone to receiving optimal osteoporosis treatment. Bone marrow density is a strong indicator for fracture events, but no bone marrow density data were included in our study. Thus, these drawbacks forced us not to analyze the fracture events in our study. Indeed, prospective clinical trials are necessary to address these possible biases and determine the cause-and-effect relationship between drug adherence of denosumab and the mortality risk.

## 5. Conclusions

In this retrospective cohort study, over a minimum of 2 years of follow-up, new denosumab users showed similar renal function status before and after denosumab treatment. New HA denosumab users may reduce all-cause mortality risk compared with LA users. A clinical trial is required to confirm our findings.

## Figures and Tables

**Figure 1 jpm-12-00185-f001:**
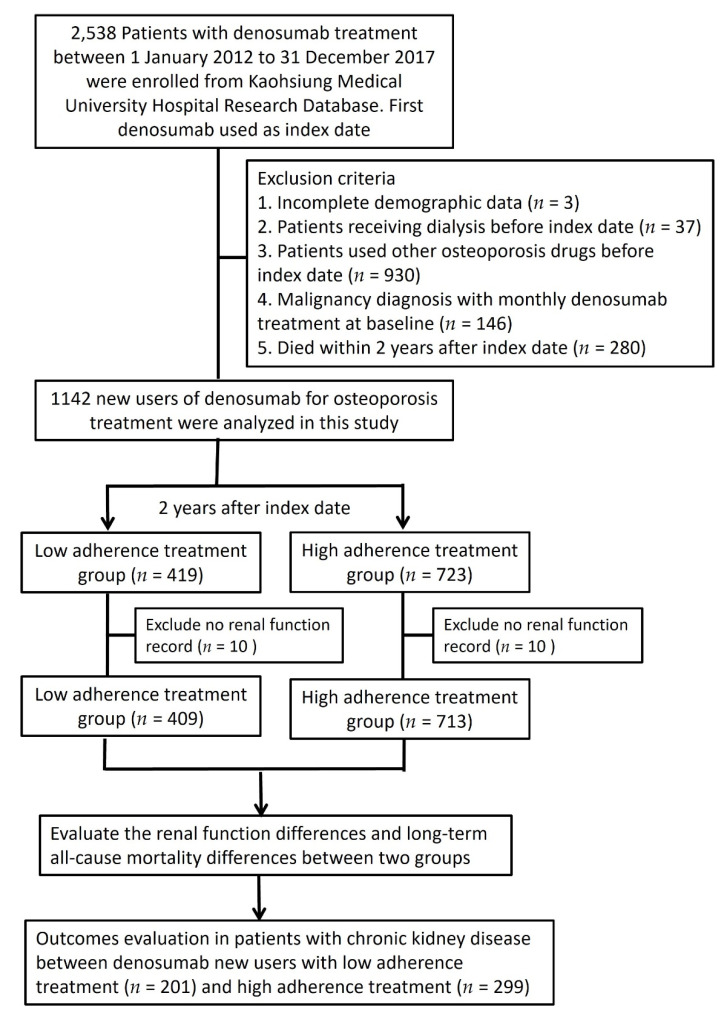
Study flowchart.

**Figure 2 jpm-12-00185-f002:**
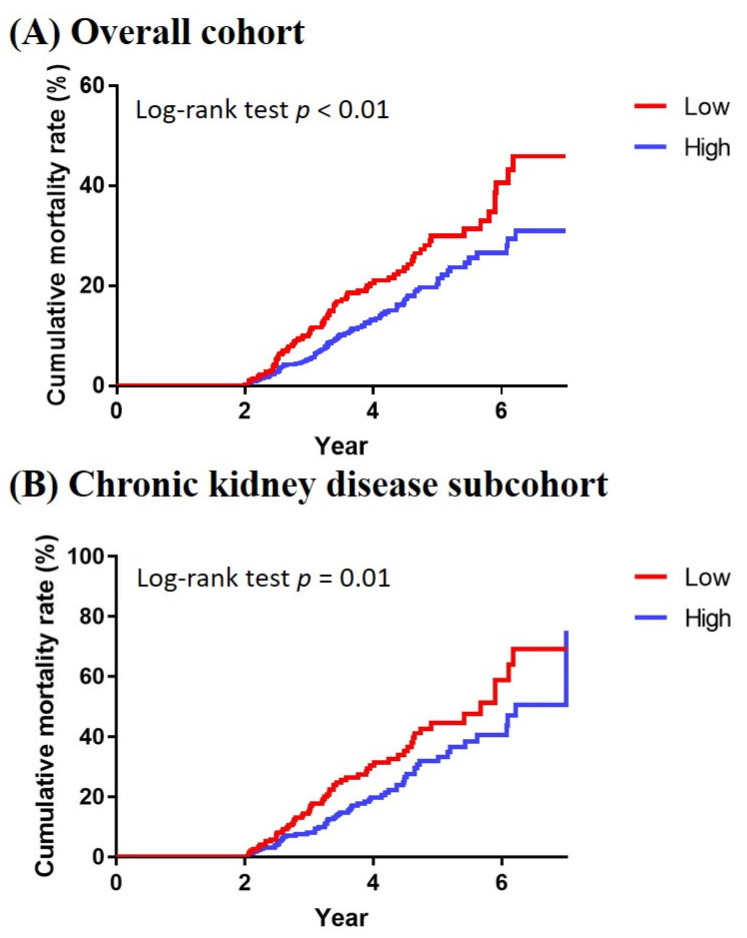
Kaplan–Meier curves for all-cause mortality, between high- and low-adherence denosumab users, in (**A**) overall cohort and (**B**) chronic kidney disease subcohort.

**Table 1 jpm-12-00185-t001:** Baseline characteristics of new denosumab users in the overall cohort and chronic kidney disease subcohort.

	Overall Cohort			Chronic Kidney Disease Subcohort		
Characteristics	Low Adherence (*n* = 409)	High Adherence (*n* = 713)	*p*-Value	Low Adherence (*n* = 201)	High Adherence (*n* = 299)	*p*-Value
Age (Mean ± SD)	75.6 ± 10.2	75.6 ± 9.5	0.95	78.0 ± 8.6	79.2 ± 8.3	0.12
Age category (*N*, %)			0.78			0.05
≤65	54 (13.2%)	86 (12.1%)		17 (8.5%)	17 (5.7%)	
65–75	119 (29.1%)	219 (30.7%)		53 (26.4%)	57 (19.1%)	
≥75	236 (57.7%)	408 (57.2%)		131 (65.2%)	225 (75.3%)	
Gender (N, %)			0.92			1.00
Male	93 (22.7%)	159 (22.3%)		48 (23.9%)	72 (24.1%)	
Female	316 (77.3%)	554 (77.7%)		153 (76.1%)	227 (75.9%)	
History of fracture (*N*, %)	289 (70.7%)	527 (73.9%)	0.27	149 (74.1%)	239 (79.9%)	0.16
Hip fracture	210 (51.3%)	348 (48.8%)	0.45	113 (56.2%)	160 (53.5%)	0.61
Non-hip fracture	113 (27.6%)	240 (33.7%)	0.04	55 (27.4%)	111 (37.1%)	0.03
Hip surgery history (*N*, %)	60 (14.7%)	107 (15.0%)	0.95	32 (15.9%)	51 (17.1%)	0.83
CCI score (Mean ± SD)	2.2 ± 2.0	2.1 ± 2.0	0.45	2.8 ± 2.1	2.8 ± 2.1	0.86
Medications						
Diabetic drugs	142 (34.7%)	208 (29.2%)	0.06	90 (44.8%)	112 (37.5%)	0.12
Antihypertensive	304 (74.3%)	491 (68.9%)	0.06	172 (85.6%)	232 (77.6%)	0.04
Lipid-lowering drugs	159 (38.9%)	310 (43.5%)	0.15	90 (44.8%)	154 (51.5%)	0.17
Anticoagulants	44 (10.8%)	70 (9.8%)	0.69	27 (13.4%)	38 (12.7%)	0.92
Diuretics	105 (25.7%)	165 (23.1%)	0.38	74 (36.8%)	106 (35.5%)	0.83
Proton pump inhibitors	86 (21.0%)	143 (20.1%)	0.76	54 (26.9%)	68 (22.7%)	0.34
NSAIDs	217 (53.1%)	429 (60.2%)	0.02	94 (46.8%)	152 (50.8%)	0.42
Corticosteroids	129 (31.5%)	231 (32.4%)	0.82	72 (35.8%)	100 (33.4%)	0.65

Abbreviations: CCI, Charlson comorbidity index; SD, standard deviation; NSAIDs, nonsteroidal anti-inflammatory drugs.

**Table 2 jpm-12-00185-t002:** Comparison of renal function in patients with denosumab treatment in the overall cohort and the chronic kidney disease subcohort.

	Low Adherence	High Adherence	*p* for Independent *t*-Test
Overall cohort	*N* = 326	*N* = 582	
Pre-eGFR (mL/min/1.73 m^2^)	66.62 ± 27.14	70.63 ± 24.70	0.02
Post-eGFR (mL/min/1.73 m^2^)	64.65 ± 27.73	70.63 ± 25.03	<0.01
*p* for paired *t*-test	<0.01	0.99	
CKD subcohort	*N* = 176	*N* = 265	
Pre-eGFR (mL/min/1.73 m^2^)	47.65 ± 20.99	49.60 ± 18.57	0.31
Post-eGFR (mL/min/1.73 m^2^)	46.35 ± 22.68	50.44 ± 20.67	0.05
*p* for paired *t*-test	0.10	0.13	

Pre-eGFR: one-year average eGFR before the first denosumab treatment (index date). Post-eGFR: one-year average eGFR after the first denosumab treatment (index date). Abbreviations: eGFR, estimated glomerular filtration rate; CKD, chronic kidney disease.

**Table 3 jpm-12-00185-t003:** Crude and adjusted hazard ratios of all-cause mortality, between high- and low-adherence denosumab users, in the overall cohort and chronic kidney disease subcohort.

	Mortality		Hazard Ratio (95% Confidence Interval)	
	No	Yes		
	*N* (%)	*N* (%)	Crude	Adjusted *
Overall cohort				
Low adherence (*n* = 409)	325 (79.5%)	84 (20.5%)	Reference	Reference
High adherence (*n* = 713)	615 (86.3%)	98 (13.7%)	0.64 (0.48−0.85)	0.64 (0.48−0.86)
CKD subcohort				
Low adherence (*n* = 201)	140 (69.7%)	61 (30.3%)	Reference	Reference
High adherence (*n* = 299)	235 (78.6%)	64 (21.4%)	0.64 (0.45−0.91)	0.61 (0.43−0.87)

* Adjusted for age, gender, and Charlson comorbidity index score. Abbreviations: CKD, chronic kidney disease.

## Data Availability

The data that support the findings of this study are not publicly available but can be accessed with permission from the Kaohsiung Medical University Hospital in Taiwan.

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
