# Peer review of "Kidney Function Change and All-Cause Mortality in Denosumab Users with and without Chronic Kidney Disease"

_jpm, 2022, doi:10.3390/jpm12020185_

Round 1
Reviewer 1 Report
Major Compulsory Revisions None.
This paper is clearly written with a clear study question, methods, results, and outcomes which are consistent with data. It offers an important addition to the literature.
Author Response
Thank you for your encouragement.
We'll work harder.
Reviewer 2 Report
This is an interesting article, an original subject with very little data world-wide.
The methodology is good, but I propose to review data about inform consent " All participants provided written informed consent to participate: (80) because it was mentioned that it is "a retrospective cohort study" (20).
Probably it should be mentioned in conclusions (241) clear data about study duration, short term study ( 2y for this time of treatment)
Congratulations nice study and I wait to see it published and indexed for meta-analysis.
Author Response
Thank you for your comments.
Point 1: The methodology is good, but I propose to review data about inform consent " All participants provided written informed consent to participate: (80) because it was mentioned that it is "a retrospective cohort study" (20).
Response 1: We've made the correction that "This study was approved by the institutional review board of KMUH (KMUHIRB-E(I)- 20210018), which waived the requirement of informed consent for this retrospective study."
Point 2: Probably it should be mentioned in conclusions (241) clear data about study duration, short term study ( 2y for this time of treatment)
Response 2: We've added the description that "In this retrospective cohort study over a minimum of 2 years follow-up , new denosumab users showed similar renal function status before and after denosumab treatment."